# Bioactive Compounds in *Moringa oleifera* Lam. Leaves Inhibit the Pro-Inflammatory Mediators in Lipopolysaccharide-Induced Human Monocyte-Derived Macrophages

**DOI:** 10.3390/molecules25010191

**Published:** 2020-01-02

**Authors:** Thitiya Luetragoon, Rungnapa Pankla Sranujit, Chanai Noysang, Yordhathai Thongsri, Pachuen Potup, Nungruthai Suphrom, Nitra Nuengchamnong, Kanchana Usuwanthim

**Affiliations:** 1Cellular and Molecular Immunology Research Unit, Faculty of Allied Health Sciences, Naresuan University, Phitsanulok 65000, Thailand; nok.hong.yok49@gmail.com (T.L.); yordhathai.k@gmail.com (Y.T.); pachuenp@nu.ac.th (P.P.); 2Thai Traditional Medicine College, Rajamangala University of Technology Thanyaburi, Pathum Thani 12130, Thailand; rungnapa_s@rmutt.ac.th (R.P.S.); chanai_n@rmutt.ac.th (C.N.); 3Department of Chemistry, Faculty of Science and Center of Excellence for Innovation in Chemistry, Naresuan University, Phitsanulok 65000, Thailand; suphrom.n1@gmail.com; 4Science Laboratory Centre, Faculty of Science, Naresuan University, Phitsanulok 65000, Thailand; Nitran@nu.ac.th

**Keywords:** *Moringa oleifera* Lam., inflammation, NF-κB pathway, monocyte-derived macrophages, active compound

## Abstract

*Moringa oleifera* (MO) is an important plant for traditional medicine. The present study aimed to identify the MO active phytochemical compounds for their ability against inflamed macrophages. An ethyl acetate extract fraction of MO was fractionation by flash column chromatography. Human macrophages were stimulated by Lipopolysaccharide and then treated with fractions of MO to examine their anti-inflammatory activity and cellular mechanism. The active fractions were analyzed by liquid chromatography with electrospray ionization quadrupole time-of-flight mass spectrometer (LC-ESI-QTOF-MS). MO treated cells showed a decreased production of pro-inflammatory mediator in response to lipopolysaccharide. This was evident at both mRNA and protein levels. The study revealed that MO suppressed mRNA expression of IL-1, IL-6, TNF-α, PTGS2, NF-κB (P50), and RelA. Furthermore, the extract effectively inhibited the expression of inflammatory mediators, including IL-6, TNF-α, and cyclooxygenase-2. Interestingly, the effect of MO inhibited phosphorylation of IκB-α and the ability to reduce expression of the nuclear factor (NF)-κB p65, suppressing its nuclear translocation. Moreover, LC-ESI-QTOF-MS analysis of the MO active fraction revealed seven compounds, namely 3,4-Methyleneazelaic acid, (2*S*)-2-phenylmethoxybutane-1,4-diol, (2*R*)-2-phenylmethoxybutane-1, 4-diol, γ-Diosphenol, 2,2,4,4-Tetramethyl-6-(1-oxobutyl)-1,3,5-cyclohexanetrione, 3-Hydroxy-β-ionone, and Tuberonic acid. Our findings highlight the ability of MO compounds to inhibit inflammation through regulation of the NF-κB pathway.

## 1. Introduction

Inflammation is a protective mechanism that is necessary in the first line of body host defense against microbial infection and injury. During inflammation, many white blood cells—such as monocytes, neutrophils, macrophages, dendritic cells, and lymphocytes—are recruited to the damaged site [1]. They can produce many cytokines—such as interleukin (IL)-1β, IL-6, IL-8, and tumor necrosis factor (TNF)-α—which promote immune cell activation and cell infiltration to the site of infection, leading to inflammation progression. However, prolonged inflammation can cause many non-communicable diseases (NCDs), including rheumatoid arthritis, diabetes, cardiovascular disease, chronic respiratory diseases, inflammatory bowel disease [2], and cancers [3]. Recently, the World Health Organization (WHO) reported that NCDs are one of the major causes of death worldwide, with an increasing proportion of premature adult deaths initiated by NCDs [4]. Nuclear factor (NF)-κB plays a key role in the regulation of inflammation by synthesis of inflammatory mediator protein and activating genes, which regulate the inflammatory response. The downstream effectors of these pathways subsequently result in the production of a variety of inflammatory mediators, such as cyclooxygenase (COX), IL-1β, IL-6, IL-8, and TNF-α to stimulate the cells and tissue responses involved in inflammation [5]. Therefore, downregulation of the NF-κB signaling pathway is one of the major targets to attenuate chronic inflammation and inflammatory diseases. The common drugs for pain and inflammation are COX inhibitors, such as nonsteroidal anti-inflammatory drugs (NSAIDs) and corticosteroids. However, long term treatment with these classical medicines may cause serious adverse effects, for example, dyspepsia, nausea, hypertension, gastrointestinal disturbances, hepatic injury, bleeding, kidney damage, respiratory depression, and cardiovascular complications [6,7]. Thus, new drugs and compounds without these effects are being investigated as alternatives for the prevention and treatment of inflammatory diseases. There are many studies related to medicinal plants and their effect on the expression of pro-inflammatory mediators, including nitric oxide (NO), nitric oxide synthase (iNOS), COX-2, IL-1β, IL-6, and TNF-α. Alternatively, these plants have been shown to increase the level of the anti-inflammatory cytokine IL-10 [8,9,10].

*Moringa oleifera* Lam. (MO) is widely cultivated in Asia and Africa, and is grown and widely used as traditional food in Thailand. Almost every part of MO provides beneficial nutrients and pharmacological properties [11]. In particular, the MO leaves have a variety of medical properties—such as hepatoprotective, antioxidant, anti-inflammatory, anti-ulcer, anti-cancer, anti-hyperglycemic, anti-bacterial, and anti-fungal activities—which can enhance the immune system [12,13]. MO leaves have been used in various in vivo studied and showed no adverse effects. Researchers found that MO dried leaf powder up to 2000 mg/kg showed no toxic in animal model without the changes in clinical signs and gross pathology. The lethal dose (LD) 50 was greater than 2000 mg/kg body weight in mice [14]. While 4.6 g per day of dehydrated MO leaf tablets used as supplement which showed anti-dyslipidemic effects and gave the overall positive impact of lipid profile in human [15]. Kushwaha et al. (2012) studied in postmenopausal women who were supplemented daily with 7 g of MO leaf powder for 3 months. The study showed that MO significant increase in serum glutathione peroxidase, superoxide dismutase, and ascorbic acid, with decrease in malondialdehyde and fasting blood glucose levels with no adverse effects [16].

In Malaysia, fraction of MO leaves have been reported to be anti-inflammatory, by inhibiting Lipopolysaccharide (LPS)-induced production of nitric oxide and the pro-inflammatory cytokines in RAW264.7 cells [17]. Another study identified that isothiocyanates, bioactive from MO leaves extract, significantly inhibited the expression of iNOS, IL-1β, and the production of NO and TNF-β [18]. Our previous study showed that an ethyl acetate MO reduced the production of pro-inflammatory cytokines, including TNF, IL-6, and IL-8 of activated human monocyte-derived macrophages (MDM) [19]. However, the effects of MO extract on the inflammatory pathway and its bioactive compounds of action in human cell still need to be investigated. Therefore, this study aimed to identify the bioactive compounds from the ethyl acetate extract of MO leaves, with in vitro cell culture of LPS-activated human MDM. Our findings clearly reveal that the compounds in Moringa leaves extract potently suppresses the targeting of the Nuclear Factor kappa B (NF-κB) pathway, leading to downregulation of inflammation processes via decreasing of pro-inflammatory mediators.

## 2. Results

### 2.1. Extraction and Fractionation of MO Leaves

A 2 kg total of dried MO powder was processed into 2 types, 44 g (2.2% yield) of crude hexane and 128 g (6.4% yield) of crude EtOAc extracts. They were subjected to test the anti-inflammatory activities compare to other solvents of MO extracts. The result showed that crude EtOAc was highest activity (Appendix A). Then crude EtOAc extracts were subjected to flash column chromatography (FCC) and received 15 fractions (Fr.1–Fr.15). All fractions were subjected to screening for the inhibition of pro-inflammatory cytokine production. Active fractions no. 6 and 12 showed anti-inflammatory activities. The fraction no. 6 was strongest activity then subjected for secondary fractionation by FCC. There were 17 sub-fractions (Fr.6.1–Fr.6.17), with only sub-fraction no.6.17 showing active anti-inflammation. Then it was partitioned by silica gel column chromatography, the result showed only one active anti-inflammation fraction (no.6.17.2). Therefore, it was subjected to compound identification by LC-ESI-QTOF-MS/MS.

### 2.2. Identification of Active Compounds from MO Extract

The active subfraction of MO leaves extract was further separated and identification of each active compound using LC-ESI-QTOF/MS. The micro-fractions of LC-MS peak were collected and tested for anti-inflammation properties. Seven compounds were tentatively identified as; compound **1**, 3,4-Methyleneazelaic acid; compound **2** and **5**, (2*S*)-2-phenylmethoxybutane-1,4-diol and (2*R*)-2-phenylmethoxybutane-1,4-diol (both showing the same *m*/*z* but a different elution order); Compound **3**, γ-Diosphenol; compound **4**, 2,2,4,4-Tetramethyl-6-(1-oxobutyl)-1,3,5-cyclohexanetrione; compound 6, 3-Hydroxy-β-ionone; and compound 7, Tuberonic acid. The compounds are shown in Figure 1A,B and the ms/ms data is shown in Table 1. While overlays of LC-MS chromatograms are shown in Appendix A.

### 2.3. Flow Cytometry Analysis of THP1, Human Monocytes, and Macrophages

Human monocytes were isolated from the buffy coat of healthy donors. Cells were maintained in completed RPMI medium and allowed to differentiate for 7 days at 37 °C, with 5% CO_2_ and media replacement every 2 days, until completely changed to human MDM. The purity of human monocytes and MDM were investigated by flow cytometry. The expression of cell surface marker CD14 and CD16 of human monocytes were 93.14% and 6.15%, respectively (Figure 2A). MDM increased the expression of CD14 and CD16 (95.72% and 96.25%), as shown in Figure 2B. THP-1 monocytic cells were differentiated into macrophages after stimulation with 5 ng/mL PMA and showed a higher expression of CD14 and CD16 (Figure 2C–E).

### 2.4. Cellular Cytotoxicity of MO Extracts and Their Fractions

After 24 h of cell incubation, the neutral red uptake cytotoxicity bioassay was performed to find out the effects of different concentrations of MO extract on MDM viability. Lethal concentration (LC) 10 and LC_50_ of each extract were analyzed from percent cell viability and using dose-response/sigmoidal curve fitting analysis. The data showed that LC_10_ of crude EtOAc extract, fraction 6, and fraction 12, were 56.98 µg/mL, 144.66 µg/mL, and 162.08 µg/mL, respectively (Figure 3A–C). These concentrations were used as the optimal dose for cell culture treatment.

### 2.5. Effect of MO Extract on Gene Expression of Pro-Inflammatory Cytokines

To investigate the effect of MO leaves extract and their fractions in LPS-stimulated macrophages, MDM were activated with LPS for 12 h and treated with MO extract and their fractions for 6 h. Then mRNA expression of IL-6, IL-10, TNF-α, COX-2, and NF-κB (P50) were evaluated by one-step quantitative real-time polymerase chain reaction. Our data showed that LPS significantly upregulated the mRNA expression of IL-1, IL-6, TNF-α, PTGS2, NF-κB (P50), and RelA, after 12 h activation in MDM compared with the normal control groups. As a positive control, dexamethasone suppressed IL-1, IL-6, TNF-α, PTGS2 (COX-2), NF-κB (P50), and RelA gene expression in LPS-stimulated macrophages. Interestingly, crude EtOAc, fractions, and sub-fractions, similarly downregulated the pro-inflammatory mRNA levels as the positive control fractions in the LPS-treated macrophages (Figure 4A). Moreover, all BPC micro-fractions (1–7) showed they were strongly affected by inhibited mRNA expression of IL-1, IL-6, TNF-α, PTGS2, NF-κB (P50), and RelA gene in LPS-treated macrophages (Figure 4B).

### 2.6. Effect of MO Extract on Pro-Inflammatory Cytokines Production

In order to investigate the changes in pro-inflammatory cytokines, IL-6, and TNF-α levels were determined after LPS- challenged MDM by sandwich ELISA kit. The results are shown in Figure 5. LPS treated cells significantly increased in both IL-6 and TNF-α cytokine productions. However, post-treatment with crude EtOAc extract, their fractions significantly reversed LPS-stimulated cytokine production to levels near the control and dexamethasone anti-inflammatory drug. Interestingly, 7 BPC micro-fractions from subfraction 6.17.2 and 3-Hydroxy-β-ionone (Santa Cruz Biotechnology, Inc., Dallas, Texas, USA) potently inhibited IL-6 and TNF-α production in inflamed cells.

### 2.7. Effects of MO Extract on LPS-Induced NF-κB and COX-2 Pathway in MDM

The nuclear factor-kappa B (NF-κB) plays a crucial role in the cell response to inflammation, particularly in macrophages. When macrophages were activated by external or internal stimuli, the activation of signal transduction cascades and transcription factor NF-κB are also affected. This can influence the production and secretion of cellular inflammatory responses, such as cyclooxygenase 2 (COX-2) and cytokines [1,20]. We investigated the effects of crude extract and fractions on LPS-stimulated cells that change the transcription factor NF-κB, phospho-IκBα, and COX-2, as they play an important role in regulating the inflammatory process. Our results of crude extract and their fractions treatment of inflamed cells demonstrated inhibition of NF-κB (p65) translocation into the nucleus, which downregulated p65, phospho-IκB-α, and COX-2 proteins levels compared with the untreated cells (Figure 6).

## 3. Discussion

Nutritional approaches in the management of inflammatory conditions include medicinal plants, which act on macrophages to prevent exacerbated inflammation. The lipopolysaccharide (LPS) endotoxin is the major component of the outer membrane of all gram-negative bacteria. LPS can bind to Toll-like receptor (TLR) 4 and activate the NF-κB signaling pathway. The activation of these cascades and transcription factors subsequently results in the releasing of proinflammatory cytokines by macrophages and circulating monocytes, resulting in a transient immune activation, which is characterized by elevated levels of TNF-α, IL-1β, and IL-6 [21]. Inflammation plays an important role in host defense against pathogen and injury, but chronic inflammation can promote the pathophysiology of many chronic diseases, such as cancer, atherosclerosis, diabetes hypertension, and cardiovascular disease [2]. Moreover, the common inflammatory drugs still provide some adverse effects, hence, compounds from natural products, which lack these side effects, have been considered as a prevention or treatment of inflammatory diseases.

As a part of this investigation on the anti-inflammatory effects of MO extract, we evaluated the levels of proinflammatory cytokines, including IL-6 and TNF-α in the supernatant of LPS-stimulated MDM. We observed a concentration-dependent decrease in the levels of IL-6 and TNF-α, while observing an increase in IL-10 level (Appendix A) in LPS-stimulated MDM, when treated with MO extract and their fraction, and the positive control drug, dexamethasone. The results demonstrated that the MO extract strongly inhibited the LPS-induced expression of IL-6 and TNF-α during inflammation while increasing anti-inflammatory cytokine IL-10. Our present findings are consistent with a previous study fraction [17], which investigated the effects of different solvent fractions, including butanol, ethyl acetate, chloroform, and hexane of MO extract. The authors observed a decrease in IL-6, TNF-α, IL-1β, and prostaglandin E2 (PGE2) production in LPS-stimulated RAW264.7 macrophages after treatment with each solvent.

Very few studies have investigated the anti-inflammatory potential of active compounds from MO leaves extract. Recently, a study identified isothiocyanates from MO leaves extract, and found that it significantly decreased gene expression and the production of inflammatory mediators in RAW macrophages [18]. We presently demonstrate that not only crude EtOAc extracts, but also their fractions, potently inhibit the response of inflamed macrophages. Moreover, MO active compounds were identified from active subfractions and found 7 active structure components, including compound **1**; 3,4-Methyleneazelaic acid, compound **2**; (2*S*)-2-phenylmethoxybutane-1,4-diol, compound **3**; γ-Diosphenol, compound **4**; 2,2,4,4-Tetramethyl-6-(1-oxobutyl)-1,3,5-cyclohexanetrione, compound **5**; (2R)-2-phenylmethoxybutane-1,4-diol, compound **6**; 3-Hydroxy-β-ionone, and compound **7**; Tuberonic acid, respectively. The presence of these compounds may be attributed to anti-inflammatory property of MO leaves extract, however no previous study has reported the anti-inflammatory activity of these compounds. 3,4-Methyleneazelaic acid was found in *Martynia annua*, an Indian plant, which has been used for self-treatment of some diseases [22]. In addition, Tuberonic acid has been found in the leaves of potatoes growing at different temperatures [23], and 3-Hydroxy-β-ionone has been isolated from *Phaseolus vulgaris* (kidney bean) [24]. Both (3*R*)-3-Hydroxy-β-ionone and (3*S*)-3-Hydroxy-β-ionone are two important intermediates in the synthesis of carotenoids, which are present in most fruits and vegetables that are commonly consumed in the United States [25].

To further confirm the mechanism related to the anti-inflammatory effect of MO extract, we evaluated the inhibitory effect of the extract on the activation of NF-κB signaling pathways and their translocation of NF-κB, from the cytoplasm to the nucleus in LPS-induced MDM. We found that phospho IκB-α, NF-κB (P65), and COX-2 were activated by the LPS, while post-treatment with MO extract inhibited phosphorylation of IκB-α and nuclear translocation of NF-κB (p65). Furthermore, the production of IL-6, TNF-α, and COX-2 were also suppressed. These data suggested that MO extract provides its anti-inflammatory activity via prevention of IκB-α degradation, and inhibition of NF-κB translocation in LPS-treated MDM, leading to down-regulation of pro-inflammatory mediators, including IL-6, TNF-α, and COX-2.

## 4. Materials and Methods

### 4.1. Extraction and Fractionation of MO Leaves

*Moringa oleifera* dried powder Lot. 5534 was obtained from Khaolaor Laboratories Co., Ltd., Samutprakan, Thailand. A 2 kg sample of powder from air-dried MO leaves, was successively extracted at room temperature with hexane (3 × 5 L) and ethyl acetate (EtOAc) (3 × 5 L). After the evaporation of solvents, the crude hexane (44 g) and crude EtOAc (128 g) extracts were obtained. The crude EtOAc extract (53 g) were subjected to FCC, eluted with a gradient system of hexane, hexane-EtOAc and EtOAc-methanol (MeOH). On the basis of their thin layer chromatography (TLC) characteristics, the fractions which contained the same major compounds were combined for further inflammatory testing. Next, the active fractions were then fractionated by silica gel column chromatography and tested for their ability against inflamed macrophages.

### 4.2. At Line-LC-ESI-QTOF-MS/MS Analysis

The active subfraction of MO leaves extract (20 mg/mL) was collected and injected to an Agilent 1260 infinity Series HPLC system (Agilent Technologies, Inc., Waldbronn, Germany) coupled to 6540 QTOF-MS spectrometers (Agilent Technologies, Inc., Singapore) with an electrospray ionization (ESI) source. The instrument was also combined with an In HPLC analysis, a mobile phase of 0.1% formic acid in water (*v*/*v*) (A) and 0.1% formic acid in acetonitrile (*v*/*v*) (B); employed in the isocratic elution with 40% solvent B for 15 min and post-run for 5 min. The sample separation was performed on the Luna C18 (2) 100°A, 4.6 × 150 mm, 5 μm (serial no. 728946-40) column (Phenomenex, California, CA, USA) at a flow rate of 0.5 mL/min, with the column temperature set at 35 °C. The eluent was split into two flows using a 9:1 ratio. The major part was collected in a 96-well plate with 30 s per well, while the minor part flowed to an ESI-QTOF-MS system. The operating parameter in MS detection was as follows: drying gas (N_2_) flow rate, 10.0 L/min; drying gas temperature 350 °C; nebulizer pressure, 30 psig; capillary, 3500 V; skimmer, 65 V; octapole RFV, 750 V; and fragmentor voltage, 100 V in positive mode. The mass range was set at *m*/*z* 100–1200 amu with a 250 ms/spectrum. The mass fragmentation was operated on auto ms/ms mode with three collision energies of 10, 20, and 40 eV, respectively. All acquisition and analysis of data was controlled by Agilent MassHunter Qualitative Analysis Software B.05.01 and Agilent MassHunter Qualitative Analysis Software B.06.0, respectively. The micro-fractions in a 96-well plate were dried using a sample concentrator (Techne, Staffordshire, UK) and kept at −20 °C before being tested.

### 4.3. Identification of Active Compounds

The samples from 96 well plate that showed bioactivity were linked to LC-ESI-QTOF/MS chromatogram by the time. The active compounds with MS and MS/MS data were tentative identification. The mass data was comparison with the previous reports and with the help of a public database; the Human Metabolomics Database (http://www.hmdb.ca; accessed on 30 December 2019), Chemspider (http://www.chemspider.com; accessed on 30 December 2019), and Metlin database (https://metlin.scripps.edu; accessed on 30 December 2019).

### 4.4. Isolation of Human Monocyte

Buffy coat were obtained from the Blood Bank, Naresuan University Hospital, Phitsanulok, Thailand. Ethics approval was obtained from the Human Ethics Committee of Naresuan University (IRB no. 1013/60). Buffy coat was diluted with a RPMI-1640 medium (Thermo Fisher Scientific, Inc., New York, NY, USA) and then overlayed on top of Lymphoprep^TM^ (STEMCELL Technologies, Singapore) for the isolation of peripheral blood mononuclear cells (PBMCs). The cells were washed once with Hank’s balanced salt solution (HBSS) and was followed by size sedimentation centrifugation using Percoll (GE Healthcare Bio-Sciences AB, Uppsala, Sweden). Human monocytes were collected and washed twice with HBSS buffer and maintained in RPMI-1640, supplemented with 10% fetal bovine serum (FBS) and 1% Antibiotic-Antimycotic (Thermo Fisher Scientific, Inc., New York, NY, USA). The cells were incubated at 37 °C with 5% CO_2_ for 2 weeks with media replacement every 2 days. Human monocyte-derived macrophages (MDM) were obtained by these processes and used in the following experiments.

### 4.5. Stimulation of THP-1 Monocytes Cells

THP-1 were obtained from ATCC and maintained in completed media RPMI 1640, supplemented with 10% FBS and 1% Antibiotic-Antimycotic (Thermo Fisher Scientific, Inc., New York, NY, USA). The differentiation of THP-1 cell line was induced by stimulation with 5 ng/mL Phorbol-12-myristate-13-acetate (PMA) (Sigma Aldrich, Missouri, MO, USA), containing RPMI for 3 days and incubating the cells in fresh completed RPMI 1640 at 37 °C with 5% CO_2_ for a further 5 days. The phenotype of THP-1 was investigated by flow cytometry.

### 4.6. Fluorescence-Activated Cell Sorting (FACS) Analysis

Cell surface markers were analysed by immunostained with PE-conjugated anti-CD14 and FITC-conjugated anti-CD16 (BD Biosciences, San Jose, CA, USA). Cells were suspended with FACs buffer solution and directly stained with fluorescence-conjugated antibody. The reaction was incubated at 4 °C for 1 h in the dark and the cells were washed twice with PBS. Cell pellet was resuspended with staining buffer and analyzed by FC 500 Flow Cytometer (Beckman Coulter, Inc., Indianapolis, IN, USA). FITC-conjugated mouse IgG1,κ isotype control and PE-conjugated mouse IgG1,κ isotype control (BioLegend, Inc., San Diego, CA, USA) were used as a negative control.

### 4.7. Cellular Cytotoxicity Assay

Neutral red uptake cytotoxicity bioassay was used to evaluate a cytotoxicity of MO extract and 3-HBI, commercial compound. MDM were seeded at the density of 5 × 10^4^ cells/well in a 96-well plate and then incubated at 37 °C in standard culture conditions for 24 h. The cells were then treated with 2-fold dilution of MO extract and incubated overnight. The cells were washed with phosphate buffered saline (PBS), then MDM was mixed with 100 µL of 100 µg/mL neutral red dye (EMD Millipore Corp., Billerica, MA, USA) in RPMI-1640 and incubated for 2 h. Removing media; cells were washed with PBS and then solubilized with 50 µL of acid-alcohol solution. The optical density was read at 545 nm by using a microplate reader (PerkinElmer, Inc., Massachusetts, MA, USA). This method has been previously described [19]. Median lethal concentration (LC_50_) of extract was calculated by dose–response relationships/sigmoidal curve fitting analysis. A ten percent lethal concentration (LC_10_) was selected as non-toxic concentration for cellular experiments.

### 4.8. In Vitro Anti-Inflammatory Analysis

Fully differentiated human macrophages were put in a 24-well plate, at the density of 2 × 10^5^ cells/well, following an incubation period of 24 h. The cells were treated with LPS of Escherichia coli O55:B5 (Sigma-Aldrich, Missouri, MO, USA) at 10 ng/mL in complete RPMI to stimulate an acute inflammatory response for 12 h. The LPS treated cells were then co-cultured with non-toxic concentration (IC_10_) of MO crude extract, their fractions and 3-HBI for 6 h in standard culture conditions at 37 °C. The control conditions used untreated cells and LPS-induced inflammation without substance added while dexamethasone was used as an anti-inflammatory positive condition.

### 4.9. Real-Time Quantitative RT-PCR

Total RNA from cells in each condition was extracted by guanidinium thiocyanate-phenol-chloroform extraction, using TRIzol^®^ reagent (Life Technologies Corporation, California, CA, USA), according to manufacturer’s protocols. The RNA of each sample was converted into their complementary DNA by reverse transcription reaction, with a Tetro cDNA Synthesis Kit (Bioline Reagents Limited, London, UK). Gene expression was evaluated with qPCR using SensiFAST SYBR No-ROX Kit (Bioline Reagents Limited, London, UK). The qPCR cycling was completed in a CFX96 Touch Real-Time PCR Detection System (Bio-Rad Laboratories, Inc., Hercules, CA, USA), set at 3 min of initial denaturation at 95 °C, 45 cycles of 10 sec denaturation at 95 °C, followed by 30 sec of annealing and elongation at 60 °C. Human beta actin gene (ACTB) was used as a housekeeping gene and RT-qPCR data was analyzed by normalized gene expression using a 2^−ΔΔCT^ method [26]. Primer sequences are shown in Table 2.

### 4.10. Enzyme-Linked Immunosorbent Assay (ELISA)

Supernatant from the cell culture of each experimental condition were collected for determination of the expression level of IL-6, TNF-α, and pro-inflammatory cytokines, by using a sandwich ELISA assay, following the manufacturer’s protocol (BioLegend, Inc., California, CA, USA). A 96-well plate was coated with capture antibody, specific to IL-6 and TNF-α at 4 °C for one night. The plate was washed four times with wash buffer between each step. The plate was then blocked with blocking solution for one hour with shaking. Supernatant from the cell culture of each condition and standard, were put into the reaction and shaken for two hours at room temperature. The plate was washed, followed by a specific binding of 100 µL of detection antibody specific to each cytokine, and incubated for one hour before washing. Avidin horseradish peroxidase-conjugated secondary antibody was then added and the plate was shaken for 30 min. The reaction was completed by adding 100 µL of freshly mixed solution of 3,3′,5,5′-Tetramethylbenzidine (TMB). After incubating for 15 min in the dark, stop solution was added. The absorbance of the reaction was measured at 450 nm using an EnSpire^®^ Multimode microplate reader (PerkinElmer, Inc., Massachusetts, MA, USA). The production of cytokines in each condition were calculated from standard curves using known concentrations of recombinant cytokines.

### 4.11. SDS-PAGE and Western Blot Analysis

The cell culture of each experimental condition was lysed by ice-cold lysis buffer in the presence of fresh protease and phosphatase inhibitor cocktails (Thermo Fisher Scientific, NY, USA), and centrifuged at 12,000 rpm for 15 min at 4 °C. The concentration of total protein, and cytoplasmic and nuclear extracts, were determined using Bradford reagent. Equal amounts of proteins were loaded on 12% SDS–polyacrylamide gel electrophoresis (PAGE), and proteins were separated according to molecular weight, and transferred to a polyvinylidene fluoride membrane (Bio-Rad Laboratories, Inc., Hercules, CA, USA). The membrane was blocked overnight at 4 °C with blocking buffer, containing 5% bovine serum albumin (Capricorn Scientific GmbH, Hesse, Germany) in Tris-buffered saline with Tween 20 (TBST) buffer. The membrane was blotted using primary antibodies specific to phospho IκBα (Invitrogen, California, CA, USA), NF-κB P65, and COX-2 (Santa Cruz, CA, USA), for 1 h at room temperature with shaking. The membrane was washed with TBST and incubated with horseradish peroxidase-conjugated goat anti-mouse IgG (H + L) secondary antibody (Thermo Fisher Scientific, Waltham, MY, USA) for an hour at room temperature. The membrane was observed by soaking in chemiluminescence substrate for 5 min and placed in a ChemiDoc XRS+ Imaging System (Bio-Rad Laboratories, Inc., Hercules, CA, USA). The chemiluminescence signal of the blotted membrane was detected by Image Studio Lite software (LI-COR Corporate, Lincoln, NE, USA).

### 4.12. Data Analysis and Statistics

All experimental conditions were triplicated to provide accurate results. Three independent batches of experiments were performed, and the results were statistically analyzed for significance. One-way analysis of variance (ANOVA) with multiple comparison method was used for data comparison among the experimental conditions.

## 5. Conclusions

In conclusion, our results strongly suggest that MO leaves extract inhibit inflammation via inactivation of NF-κB, blocking both IκB-α degradation and nuclear translocation of p65. The NF-κB subunits thereby inhibit the binding of NF-κB to its target DNA, resulting in downregulation of the pro-inflammatory mediators, IL-6, TNF-α, and COX-2 (Figure 7). Moreover, we found that MO are strong radical scavengers and can be considered as good sources of natural antioxidants for the treatment of inflammation-associated diseases. However, further investigations are needed—both in vivo and clinical trials—to confirm the anti-inflammatory effects of MO extract.

LPS binding to Toll-like receptor 4 (TLR4) leading to activation of NF-κB pathway and phosphorylation of the inhibitor of kappa B (IκB-α) by IκB kinase (IKK), resulting in degradation of IκB-α. The free NF-κB subunits translocate from the cytoplasm into the nucleus where it induces the transcription of genes encoding for pro-inflammatory mediators, such as TNF-α, IL-6, and COX-2. While MO extract inhibits the phosphorylation of IκB-α, which prevents the nuclear translocation of NF-κB and suppresses the expression of inflammatory mediators.

## Figures and Tables

**Figure 1 molecules-25-00191-f001:**
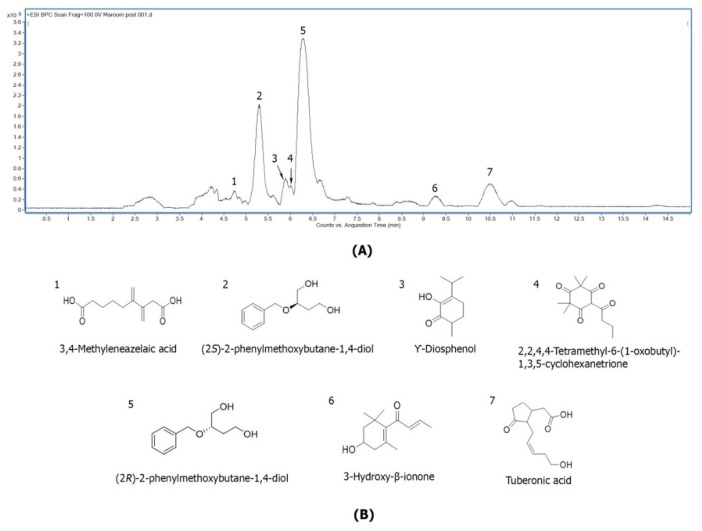
Base peak chromatogram of MO subfraction at concentration 20 mg/mL in at line system. (**A**) LC-MS base peak chromatogram (BPC) of subfraction 6.17.2 with active compounds no.**1**–**7** and (**B**) the tentative structure.

**Figure 2 molecules-25-00191-f002:**
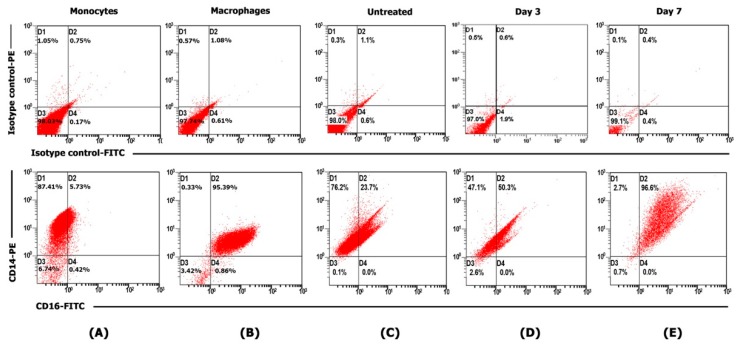
Dot plot of representative flow cytometry profiles on gated monocytes, macrophages, and THP-1 cell lines. (**A**,**B**) Purification of monocytes and macrophages, (**C**–**E**) Differential response to PMA by THP-1 in 7 days. Untreated: THP-1 in RPMI medium; Day 3: treated THP1 with PMA 5 ng/mL at day 3; Day 7: treated THP1 with PMA 5 ng/mL at day 7.

**Figure 3 molecules-25-00191-f003:**
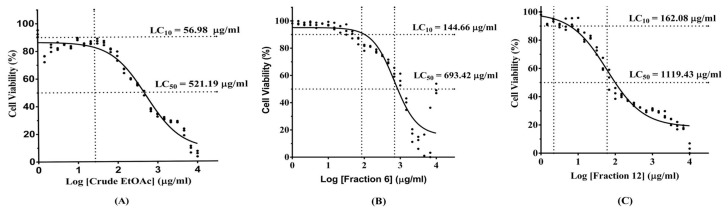
Effect of various MO leaves extract concentrations in the neutral red assays. (**A**) Dose–response curve showing 10 lethal concentration (LC_10_) and median percent lethal concentration (LC_50_) of crude EtOAc extract, (**B**) Fraction 6, (**C**) Fraction 12.

**Figure 4 molecules-25-00191-f004:**
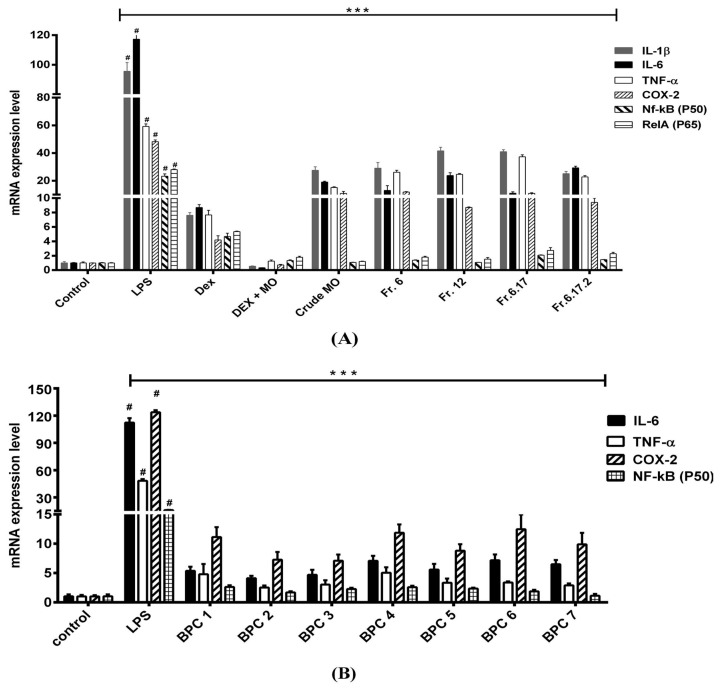
Effect of MO leaves extract on mRNA expression of pro-inflammatory cytokines levels in MDM. The cytokine mRNA expression of treated MO leaves extract and their fractions (**A**) and mRNA expression of 7 micro-fractions collected from a line LC-MS/MS of subfraction 6.17.2 (**B**). Data are presented as means ± SEM. # *p* < 0.001 compared to control, *** *p* < 0.001, compared to LPS. Control: Untreated MDM; LPS: Lipopolysaccharide-stimulated MDM; Dex: Dexamethasone; Dex + MO: Combination of Dexamethasone and crude EtOAc; crude MO: ethyl acetate MO; Fr6: Fraction 6; Fr12: Fraction 12; Fr.6.17: Fraction 6.17; Fr.6.17.2: Fraction 6.17.2.; BPC: base peak chromatogram.

**Figure 5 molecules-25-00191-f005:**
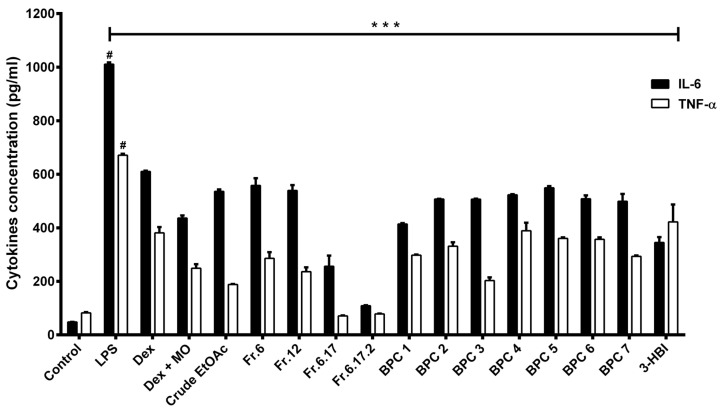
Effect of MO leaves extract on pro-inflammatory cytokines levels in LPS-induced MDM. Data are presented as means ± SEM. # *p* < 0.001 compared to control, *** *p* < 0.001, compared to LPS. Control: Untreated MDM; LPS: Lipopolysaccharide-stimulated MDM; Dex: Dexamethasone; Dex + MO: combination of Dexamethasone and crude EtOAc; Crude EtOAc: Crude ethyl acetate; Fr6: Fraction 6; Fr12: Fraction 12; Fr.6.17.2: Fraction 6.17.2.; BPC: base peak chromatogram; 3-HBI: 3-Hydroxy-β-ionone.

**Figure 6 molecules-25-00191-f006:**
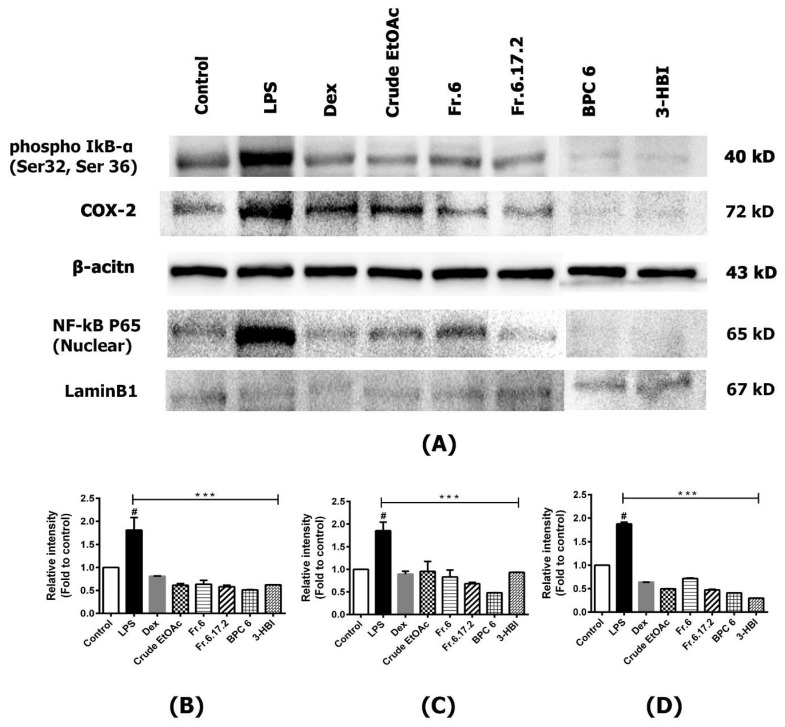
Effect of MO extract on the protein expressions in MDM. (**A**) Total protein and nuclear protein levels by western blotting using specific antibodies, including NF-κB P65, phospho-IκBα, and cox-2; β-actin was used as a loading control. (**B**) Relative protein levels of phospho-IκBα, (**C**) COX-2, and (**D**) NF-κB P65 (Nuclear), were quantified by scanning densitometry and normalized to control. Data are presented as means ± SEM. # *p* < 0.001 compared to control, *** *p* < 0.001, compared to LPS. Control: Untreated MDM; LPS: Lipopolysaccharide-stimulated MDM; Dex: Dexamethasone; Dex + MO: combination of Dexamethasone and crude EtOAc; Crude EtOAc: Crude ethyl acetate; Fr6: Fraction 6; Fr6.17.2: Fraction 6.17.2; BPC 6: base peak chromatogram 6; 3-HBI: 3-Hydroxy-β-ionone.

**Figure 7 molecules-25-00191-f007:**
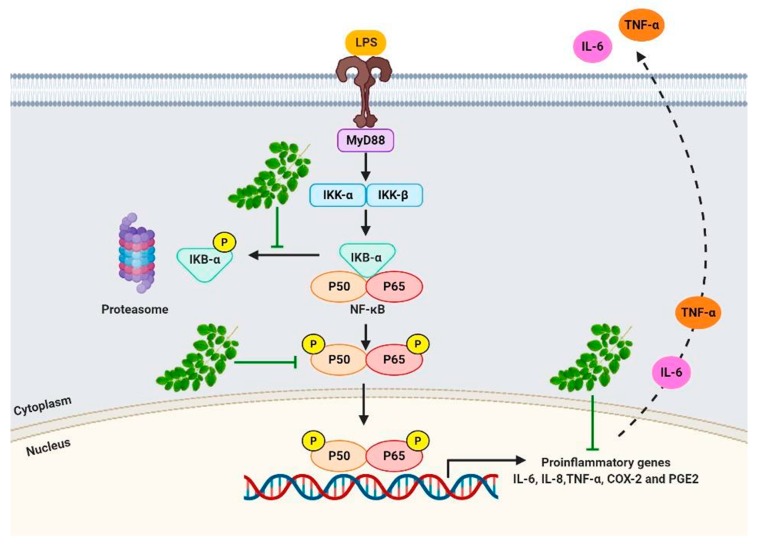
Mechanism of anti-inflammation of MO in LPS-induced macrophages.

**Table 1 molecules-25-00191-t001:** MS data of (+) ESI-QTOF-MS of MO subfraction and tentative identification

RT	*m*/*z*[M + H]+	MS/MS	Tentative Identification	Formula	Error (ppm)
4.73	213.1111	195.10, 177.09, 151.07, 149.09, 121.15	3,4-Methyleneazelaic acid	C_11_H_16_O_4_	4.86
5.38	197.1183	179.10, 161.09, 133.09, 107.08	(2*S*)-2-phenylmethoxybutane-1,4-diol	C_11_H_16_O_3_	−5.47
5.88	169.1229	127.11, 111.07, 77.03, 55.05, 57.06	γ-Diosphenol	C_10_H_16_O_2_	−3.51
6.02	253.1444	235.13, 217.12, 202.09, 175.11, 151.07, 119.08	2,2,4,4-Tetramethyl-6-(1-oxobutyl)-1,3,5-cyclohexanetrione	C_14_H_20_O_4_	−3.81
6.25	197.1167	179.10, 161.09, 133.10, 107.08	(2*R*)-2-phenylmethoxybutane-1,4-diol	C_11_H_16_O_3_	2.64
9.3	209.145	191.13, 173.12, 151.10, 139.10, 121.09, 107.08	3-Hydroxy-β-ionone	C_13_H_20_O_2_	−3.79
10.43	227.1281	195.09, 153.08, 125.09, 85.09, 68.99	Tuberonic acid	C_12_H_18_O_4_	−1.38

**Table 2 molecules-25-00191-t002:** Primers sequences use in real-time PCR analysis.

Gene	Sequence (5′ → 3′)		Tm (°C)	Amplicon Size (Base Pair)
***IL-1B***	*Homo sapiens interleukin 1, beta (IL1B), mRNA*			
	F:	AGCTACGAATCTCCGACCAC	61.1	186
	R:	CGTTATCCCATGTGTCGAAGAA	60.1	
***IL6***	*Homo sapiens interleukin 6 (interferon, beta 2), mRNA*			
	F:	ACTCACCTCTTCAGAACGAATTG	60.2	149
	R:	CCATCTTTGGAAGGTTCAGGTTG	61.3	
***IL-10***	*Homo sapiens interleukin 10 (IL10), mRNA*			
	F:	TCAAGGCGCATGTGAACTCC	60.8	176
	R:	GATGTCAAACTCACTCATGGCT	60.2	
***TNF***	*Homo sapiens tumor necrosis factor, mRNA*			
	F:	CCTCTCTCTAATCAGCCCTCTG	60.8	220
	R:	GAGGACCTGGGAGTAGATGAG	60.2	
***RelA***	*Homo sapiens v-rel reticuloendotheliosis viral oncogene* *homolog A (avian), transcript variant 2, mRNA*			
	F:	ATGTGGAGATCATTGAGCAGC	60.2	151
	R:	CCTGGTCCTGTGTAGCCATT	61.3	
***PTGS2***	*Homo sapiens prostaglandin-endoperoxide synthase 2* *(prostaglandin G/H synthase and cyclooxygenase)* *(PTGS2), mRNA.*			
	F:	CTGGCGCTCAGCCATACAG	62.8	94
	R:	CGCACTTATACTGGTCAAATCCC	61.0	
***ACTB:***	*Beta actin; beta cytoskeletal actin [Homo sapiens]*			
	F:	CATGTACGTTGCTATCCAGGC	60.8	250
	R:	CTCCTTAATGTCACGCACGAT	60.2	

Primer sets for RT-qPCR were select from PrimerBank (https://pga.mgh.harvard.edu/primerbank/; accessed on 30 December 2019). Primer should be between 18 to 25 nucleotides in length and melting temperature (Tm) of the hairpin should range between 55 °C and 65 °C.

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
