# Peer review of "Bioactive Compounds in Moringa oleifera Lam. Leaves Inhibit the Pro-Inflammatory Mediators in Lipopolysaccharide-Induced Human Monocyte-Derived Macrophages"

_molecules, 2020, doi:10.3390/molecules25010191_

Round 1

Reviewer 1 Report

Comments for the paper “Bioactive Compounds in Moringa oleifera Lam. Leaf Inhibit the Pro-inflammatory Mediators in Lipopolysaccharide-Induced Human Monocyte- Derived Macrophages”.

The authors revealed the effect of compounds in Moringa oleifera Lam. leaf on LPS-stimulated macrophages.

Major Comments

1) The novelty of this study is not strong. The authors mentioned previously similar studies in the Introduction section. These previous studies all suggested the results obtained in this manuscript. Namely, the novelty of the present manuscript is limited.

2) As the authors understand, statistical analysis is very important to assess the research results. In the Statistical analysis, the authors used paired-sample t-test. I think that the authors should use the multiple comparison method following one-way analysis of variance. I did not make a judgment about the importance of this manuscript in this version.

3) Fig.2. Fr6 is more strong activity than crude-EtoAc. Is this due to the concentration of the compounds in Fr6 and crude-EtoAc? Or does crude-EtoAc contain a substance that suppresses activity?

4) Please explain the reason and purpose of experiment in Fig.3. Please explain about CD14 and CD16.

5) Fig. 5 and Fig. 6. Please indicate the concentration used. Are you sure that the cell is not dead at that concentration?

6) Fig. 5 and Fig. 6. Although the seven compounds shown in Fig.1 are quite different in their structural formulas, why did these compounds suppress cytokine production to the same extent? Have you examined the permeability of these seven compounds to cell membranes?

7) Fig.7. The authors should examine seven compounds.

Minor Comments

1) P.2 There are two abbreviations for IL.

2) P.2 There are two abbreviations for TNF.

3) P.2 The abbreviation of LPS is wrong.

4) There are three abbreviations for CPD.

Author Response

Dear Reviewer 1

Please see the attachment of our response.

Best regards

Kanchana Usuwanthim

Correspondence

Reviewer 2 Report

Although the paper provides interesting results, I have major comments and concerns that should be addressed and clarified.

Major concerns:

1. Did the authors test different solvents than ethyl acetate?. It has been stated that the authors, in a previous work examined an ethyl acetate MO, which showed a reduction of the production of pro-inflammatory cytokines, including TNF, IL-, and IL-8 of activated human monocyte-derived macrophages (MDM). It's obvious that the authors wanted to follow up their previous study, but in my opinion, a comparison between several solvents could give a better overview of the best solvent that could be used to extract bioactive fractions/molecules with anti-inflammatory properties. Please, clarify this point in the manuscript.

2. Are the authors sure that only 7 compounds were found in the fraction of MO and no other compounds or isomers were detected? Please, provide the HPLC chromatogram of the separated compounds in the MO fraction. This will enhance the accuracy of your analyses.

3. Since the identified compounds are commercially available, why did not the authors examine them against all biological assays performed in the work? In my opinion, this could enhance the significance of the work.

Minor comments

1. Introduction

Since Moringa oleifera Lam. is widely cultivated in Asia and Africa, and is grown and widely used as a traditional food in Thailand, it would be better to provide a background on the safety profile of this plant.

2. Results

Table 1. all numbers in the molecular formula of identified compounds should be subscripted.

3.  Discussion

Line 330. Phaseolus vulgaris should be written in italic. I recommend the authors to pay attention to all scientific nomenclatures and to check them correctly.

4. Materials and methods

Table 2. Based on which criteria the used primers were chosen? Please, clarify this point below the table.

4.14. Data Analysis and Statistics

Did the authors use some post-hoc comparison tests along with ANOVA to assess the differences between treatments with test samples and positive control? Please clarify.

Author Response

Dear Reviewer 2

Please see the attachment of our response.

Best regards

Kanchana Usuwanthim

Correspondence

Round 2

Reviewer 2 Report

The paper has been significantly improved. 

Author Response

Dear Reviewer

We add Western blot result of BPC 6 and 3-HBI in Figure 6 and check English expression in manuscript.

Thank you for your comments and suggestions.

Best regards,

Kanchana Usuwanthim